

# Testing the influence of light on nitrite cycling in the eastern tropical North Pacific

Nicole M. Travis[1], Colette L. Kelly[1], Karen L. Casciotti[1,2]

[1]Earth System Science, Stanford University, Stanford CA 94305 USA
[2]Oceans Department, Stanford University, Stanford CA 94305 USA

*Correspondence to: Nicole M. Travis (ntravis@stanford.edu)*

**Abstract.** Light is considered a strong controlling factor on nitrification rates in the surface ocean. Previous work has shown that ammonia oxidation and nitrite oxidation may be inhibited by high light levels, yet active nitrification has been measured in the sunlit surface ocean. While it is known that photosynthetically active radiation (PAR) influences microbial nitrite production and consumption, the level of inhibition of nitrification is variable across datasets. Additionally, phytoplankton have light-dependent mechanisms for nitrite production and consumption that co-occur with nitrification around the depths of the primary nitrite maximum (PNM). In this work, we experimentally determined the direct influence of light level on net nitrite production, including all major nitrite cycling processes (ammonia oxidation, nitrite oxidation, nitrate reduction, nitrite uptake) in microbial communities collected from the base of the euphotic zone. We found that although ammonia oxidation was inhibited at the depth of the PNM and was further inhibited by increasing light at all stations, it remained the dominant nitrite production process at most stations and treatments, even up to 25% surface PAR. Nitrate addition did not enhance ammonia oxidation in our experiments, but may have increased nitrate and nitrite uptake at a coastal station. In contrast to ammonia oxidation, nitrite oxidation was not clearly inhibited by light, and sometimes even increased at higher light levels. Thus, accumulation of nitrite at the PNM may be modulated by changes in light, but light perturbations did not exclude nitrification from the surface ocean. Nitrite uptake and nitrate reduction were both enhanced in high light treatments relative to low light, and in some cases showed high rates in the dark. Overall, net nitrite production rates of PNM communities were highest in the dark treatments.

## 1 Introduction

Accumulation of nitrite in the surface ocean in the primary nitrite maximum (PNM) is controlled by four dominant microbial processes, including ammonia oxidation, nitrite oxidation, nitrate reduction and nitrite uptake. The nitrification processes (ammonia oxidation and nitrite oxidation) are performed by specialized archaeal and bacterial cells, while nitrate reduction and nitrite uptake are largely light-dependent phytoplankton processes. Activity from these microbial groups has been measured near the PNM feature, but it is unclear what environmental conditions control the relative rates of these four microbial processes and the resulting concentrations of nitrite.

Light is an environmental parameter often suggested to control nitrification rates. Observed nitrification rates in ocean profiles are typically low in surface waters and increase to maximum rates at the base of the euphotic zone (Ward, 1985). These correlative



patterns across depth suggest that nitrification is inhibited by light. In the eastern tropical North Pacific ocean (ETNP), paired
nitrification measurements (ammonia oxidation and nitrite oxidation) showed patterns where low light levels (<5% surface PAR)
corresponded to the majority of high nitrification rates (>10 nM d$^{-1}$) in the ETNP, although active nitrification was still occasionally
measured in samples with light levels >10% surface PAR (Travis et al., 2023). 06/04/2023 07:37:00Other work in the ETNP has
shown ammonia oxidation rates are excluded from light levels above ~1-5% of surface PAR (Beman et al., 2012), while data from
Monterey Bay, CA showed rates up to 35 nM d$^{-1}$ even at >90% surface PAR (Ward, 2005). Direct light inhibition has also been
confirmed in cultured marine ammonia-oxidizing archaea (Merbt et al., 2012) and the marine nitrite-oxidizing bacteria *Nitrococcus
mobilis* and *Nitrobacter sp.* (Guerrero and Jones, 1996a, b).

Differential light inhibition is a common theory posited to cause imbalance in the two steps of nitrification resulting in accumulation
of nitrite (Brandhorst, 1958; Francis et al., 2005; Guerrero and Jones, 1996a, b; Hooper and Terry, 1974; Lomas and Lipschultz,
2006; Mackey et al., 2011; Meeder et al., 2012; Merbt et al., 2012; Olson, 1981a). In order for differential light inhibition of
nitrifiers to cause an imbalance leading to nitrite accumulation, nitrite-oxidizing bacteria would have to be more light-sensitive
than ammonia oxidizers. Prior studies have shown that nitrifiers are light sensitive, but there is a lack of consensus on whether
nitrite oxidizers (Olson, 1981a) or ammonia oxidizers are more photosensitive (Guerrero and Jones, 1996a; Hooper and Terry,
1974; Horrigan and Springer, 1990). At the same time, measurements of ammonia oxidation and nitrite oxidation in the field are
rarely in balance. It is also likely that light sensitivities are modulated by other environmental conditions; for example, substrate
replete conditions and optimal temperatures are known to mitigate light sensitivity in some microbes. Recent work has suggested
that instead of direct light inhibition, observed decreases in ammonia oxidation rate in near-surface waters could be attributable to
increased competition with phytoplankton for substrates (Smith et al., 2014; Wan et al., 2018). This competition for ammonium
has been postulated to be modulated by nitrate availability and light, where increased light causes increased ammonium affinity in
phytoplankton and simultaneous declines in ammonium affinity for ammonia oxidation, giving phytoplankton a distinct advantage,
especially in low-nutrient environments (Xu et al., 2019).

Phytoplankton are also influenced by light, with enhanced growth and N uptake at higher light levels. They have been observed to
release nitrite under variable light and nutrient conditions (Collos, 1998; Kiefer et al., 1976; Lomas and Glibert, 2000; Lomas et
al., 1999; Sciandra and Amara, 1994; Vaccaro and Ryther, 1960; Wada and Hattori, 1971). The physiological cause of nitrite
release from phytoplankton is unclear, but it has been linked to nitrate uptake activity in dark and low light conditions and was
attributed to incomplete nitrate assimilation (Vaccaro and Ryther, 1960; Kiefer et al., 1976). Other studies suggest that sporadic
high light events stimulate excess nitrate reduction as a photosynthetic energy dissipation pathway (Lomas and Glibert, 1999;
Lomas et al., 1999).

Many phytoplankton are also capable of nitrite uptake, although low availability of nitrite in the field can make using nitrite less
favorable than using nitrate. Nitrate uptake rates are generally higher than those of nitrite uptake for many phytoplankton when
both substrates are available (Collos, 1998). Nitrite reduction is an energy intensive process, and adequate light availability
typically controls nitrite uptake (Collos and Berges, 2003; Berges, 1997; Berges and Harrison, 1995; Berges et al., 1995; Hattori
and Wada, 1971; Lomas and Glibert, 2000). Wada and Hattori (1971) measured nitrite uptake in dark and light bottles in the ETNP,
and confirmed that field assemblages take up more nitrite under higher light conditions. Thus, photosynthetic microbes are a
relatively cryptic source of nitrite to the PNM because they are capable of both nitrite production and consumption. This dual





function as a source and sink term for nitrite allows phytoplankton to control nitrite accumulation on their own, or to become a competitor for the substrates required in nitrification (Smith et al., 2014; Wan et al., 2018). It can be difficult to discern what
controls whether phytoplankton communities act as a net source or sink of nitrite in the field.

Both the uncoupling of the two steps of nitrification and nitrite release by phytoplankton (via nitrate reduction) have been used independently to explain PNM formation. However, it is likely that these processes co-occur (Wan et al., 2021). The relative rates of each process are controlled by the ecophysiological response of each microbial group to environmental conditions, leading to
80 dynamic changes in net accumulation of nitrite (Carlucci et al., 1970). Few direct measurements attempt to separate all of the relevant, overlapping nitrite consumption and production rates in the field (Kiefer et al., 1976; Olson, 1981b; Travis et al., 2023). Experimental manipulations of light and nitrate availability are needed to understand the controls that regulate the balances between source and sink processes and between phytoplankton and nitrifier processes, and to separate the effects of microbial community composition from direct impacts of light and nutrients on the measured rates of these essential reactions.

In this study, we used natural microbial populations collected from PNM depths to experimentally determine the influence of light level and nitrate concentration on the relative rates of the four dominant microbial processes influencing nitrite accumulation. Our experimental manipulations provided insight into the physiological responses of the community that are distinct from conclusions obtained from the natural distributions of instantaneous rates across environmental gradients. Instantaneous rate distributions are
90 reflections of the ambient environmental conditions (including light level), in addition to the natural community composition, whereas experimental manipulations illustrate responses of a specific community to environmental perturbations. We hypothesized that increased light intensity would lead to a shift towards higher phytoplankton activity and lower nitrification rates. We expected net nitrite production to decline at higher light levels, with nitrate reduction becoming a larger proportion of net production as ammonia oxidation rates declined. We also expected nitrate addition to cause an increase in phytoplankton nitrate uptake and a
95 corresponding increase in ammonia oxidation rates through alleviation of substrate competition (Wan et al., 2021).

## 2 Methods

### 2.1 Site description and experimental design

Data were collected aboard the *R/V Sally Ride* (SR1805) from March to April 2018 and aboard the *R/V Falkor* (FK180624) from
100 June to July 2018 in the Eastern Tropical North Pacific (ETNP). The SR1805 cruise transect spanned a straight path from near the western edge of the ETNP oxygen deficient zone (ODZ) at an offshore process station (PS1, 10°N, 113°W) towards a coastal process station (PS3, 17.7°N, 102.4°W) where experimental rates measurements were conducted. An additional process station was occupied near the geographic center of the oxygen deficient zone (PS2, 15.8°N, 105°W) (Fig. 1). While this study focused on euphotic zone processes, the region is underlain by a functionally anoxic zone, which was the focus of related studies (eg. Kelly et
al., 2021; Sun et al., 2021; Frey et al., 2023). Hydrographic data were collected at each station using a Seabird SBE 911+ CTD package mounted either on a 12 or 24 Niskin bottle rosette (Temperature, Salinity, Pressure). Fluorescence data and photosynthetically active radiation (PAR) were collected using sensors on the 12-bottle rosette at each of the stations (PS1, PS2 and PS3). The FK180624 cruise transect occupied stations along 14°N latitude from ~102°W to ~116°W (Fig. 1). Experimental rate measurements were made at Station 2 (14°N, 103°W) and Station 9 (14°N, 110°W). Hydrographic data were collected at each





station using Seabird SBE 911+ CTD package mounted on a 12 Niskin bottle rosette. A 150-mL polycarbonate (PC) bottle was triple-rinsed and used to collect discrete samples for ambient source water dissolved inorganic nitrogen (DIN, including $NO_3^-$, $NO_2^-$, and $NH_4^+$) concentration measurements at each station and depth.

To determine the influence of light and nitrate concentration on microbial nitrite cycling, source water was collected from the lower slope of the PNM at experimental stations during a pre-dawn cast. Where available, nitrite concentration data from a previous cast were used to target the depth of the PNM at a given station; otherwise, the depth of the lower slope of the chlorophyll maximum guided water collection depths. Each source water community was incubated at four light levels, with and without an additional 20 μM nitrate ($KNO_3$). Low-light (LL), medium-light (ML) and high-light (HL) treatments (approximately 1%, 4% and 20% surface irradiance, sPAR) were achieved in flow-through seawater incubators with layered window-screening designed to maintain irradiance at the desired levels. A dark (DK) treatment was achieved using brown HDPE bottles and incubating in the 1% sPAR tank. Light levels in each incubator were directly measured during the cruise using a LicoR submersible PAR meter or a submerged HOBO LUX data logger. The deck-board incubation tank was continuously fed with surface seawater to maintain consistent temperature.

**2.2 Rate measurements**

Samples for rate measurements were collected directly from the Niskin bottles by triple rinsing and filling replicate experimental containers. Experimental bottles included 500-mL high-density brown polyethylene (HDPE) bottles for dark-incubated treatments, and corresponding 500-mL clear polycarbonate (PC) bottles for light-incubated experiments. [15]N tracer appropriate for each process was then added to replicate incubation bottles: ammonia oxidation (98.8 *atm%* [15]N-NH₄Cl), nitrate reduction (98.8 *atm%* [15]N-KNO₃), and nitrite oxidation/uptake (98.8 *atm%* [15]N-NaNO₂) (Sigma-Aldrich). At the coastal station PS3, uptake of ammonium and nitrate were also measured using [15]N-NH₄Cl and [15]N-KNO₃ tracers, respectively. The appropriate [15]N tracer solution was added at the start of each incubation to reach 200 nM [15]N for all experiments and gently mixed. For experimental treatments with added nitrate, 20 μM of unlabeled KNO₃ solution was added to the incubation bottle.

Rate estimates are susceptible to stimulation from the [15]N additions used to track transformation of substrate into the product pools. Tracer experiments often aim for 10% [15]N addition to minimize rate stimulation from the added nitrogen, but this method relies on the substrate pool being large enough to consistently add [15]N at ~10% levels, which is impractical in regions where nitrogen concentrations are highly variable or very low. The determination of rates also depends on the assumptions that the labeled fraction of source DIN remains constant, and only a small percentage of the [15]N-labeled source pool ends up in the product. If consumption of the source DIN is complete (i.e., 100% of [15]N spike ends in the product), this can lead to an underestimate of the rate. Dilution of the source DIN pool during the course of the experiment (e.g., regeneration of ammonium from grazers) will also lead to underestimation of rates, especially over the course of longer incubation times. In this study, the addition of [15]N at a uniform level of 200 nM across all experiments was the most feasible design for implementation across multiple cruises, stations, depths, and DIN sources where nitrogen concentrations were variable. Given this, our [15]N spikes ranged from <1% to >90% of the source nitrogen pool, and have the potential of stimulating the measured rates, especially in the higher % enrichment experiments. The potential enhancement of rates does not preclude comparison of light treatment effects.



After the $^{15}$N spike was added, a subsample was immediately filtered (Sterivex, 0.22 μm pore size syringe filter) into a 60-mL HDPE bottle and frozen at -20°C to represent initial conditions for later isotope analysis and nitrate concentration measurements

upon return to Stanford University. An aliquot of each initial sample was also analyzed shipboard for ammonium and nitrite concentrations to confirm 200 nM $^{15}$N additions. At timepoints approximately 8-hours, 16-hours and 24-hours from initial spike time, a subsample was Sterivex-filtered (0.22 μm pore size) into a 60-mL HDPE bottle and frozen for later isotope analyses and rate calculations. At the 24-hour time point, the remaining incubation water was combined from experimental replicates (to maximize particulate nitrogen content) and vacuum-filtered onto a pre-combusted (450°C for > 4 h) GF/F filter (0.7 μm nominal

pore size). Filters were folded, placed in cryovials and stored at -80°C for later analysis of particulate $^{15}$N and DIN uptake rate calculations. Between experiments, bottles were acid washed and re-used for experiments with the same $^{15}$N-DIN type.

### 2.3 Chemical concentrations

Nitrite concentrations were measured ship-board with a spectrometer using colorimetric methods and calibrated with a standard curve bracketing the expected nitrite concentrations of samples (Strickland and Parsons, 1972). Briefly, 5 ml of sample or standard was reacted with 200 μl each of sulfanilamide (SAN) and N-(1-Naphthyl)ethylenediamine (NED) reagents and absorbance at 543 nm was measured after 10 min of color development. The limit of detection was ~200 nM. Ammonium concentrations were measured shipboard by fluorometry using an adapted *o*-phthalaldehyde (OPA) method (Holmes et al., 1999, as modified in Santoro

201X). Standard curves were made by standard addition to a seawater matrix, with water collected from below the euphotic zone. Samples and standards were incubated using OPA reagent for ~8 hours before measurement. The limit of detection for this method was 30 nM. Nitrate concentrations were measured against a bracketing standard curve using a WestCo SmartChem 200 Discrete Analyzer at Stanford University, with a detection limit of 85 nM and precision of 0.6 μM (Miller and Miller, 1988; Rajaković et al., 2012).


### 2.4 Isotopic analyses

For estimates of ammonia oxidation, nitrite oxidation and nitrate reduction rates, the 0- and 8-h timepoints from each incubation were analyzed for $^{15}$N enrichment in the product pools. Product DIN was converted to nitrous oxide either by bacterial conversion

using the denitrifier method (Sigman et al., 2001; McIlvin and Casciotti, 2011) or chemical conversion using the azide method (McIlvin and Altabet, 2005).

**Table 1. Tracer additions and preparation process for rate measurements**

| Process | $^{15}$N-labeled reactant | $^{15}$N-labeled product | Prep Method |
|---|---|---|---|
| Ammonia Oxidation | NH$_4$Cl | NO$_X$ | Denitrifier method |
| Nitrite Oxidation | NaNO$_2$ | NO$_3^-$ | Sulfamic treated + denitrifier method |
| Nitrate Reduction | KNO$_3$ | NO$_2^-$ | Azide method w/ carrier |
| Nitrite Uptake | NaNO$_2$ | Particulate N | Dry and pack in tin |

Both ammonia oxidation and nitrite oxidation measurements utilized the denitrifier method to quantify $^{15}$NO$_X^-$ in the product pool. For nitrite oxidation measurements, NO$_2^-$ in $^{15}$NO$_2$-spiked samples was removed by pre-treatment with 4% sulfamic acid solution



and 2M sodium hydroxide prior to conversion of the remaining nitrate to $N_2O$ via bacterial denitrification, resulting in analysis of nitrate-derived $N_2O$ only (Granger and Sigman, 2009). Nitrate reduction measurements utilized chemical conversion of product $NO_2^-$ to $N_2O$ with azide (McIlvin and Altabet, 2005). Briefly, after removal of background $N_2O$ by purging with $N_2$ gas, samples

were treated with 2 M sodium azide solution in 20% acetic acid for ~30 min followed by neutralization with 6 M sodium hydroxide. The nitrite product pool in nitrate reduction samples was often highly enriched in $^{15}N$ due to low ambient nitrite concentrations; therefore, additional carrier $NaNO_2$ was added prior to isotopic analysis (25 μl of 200 μM $NaNO_2$ in 10 ml sample). After analysis, the isotopic composition of the carrier-diluted samples was calculated by subtracting out the known isotopic value and concentration of the added $NaNO_2$ carrier using mass balance.


Isotopic enrichment of the resulting $N_2O$ in all cases was determined using a Thermo-Finnigan Delta$^{PLUS}$XP or Delta V$^{PLUS}$ isotope ratio mass spectrometer connected to a custom-built cryogenic purge and trap system with autosampler (PAL) (McIlvin and Casciotti, 2011). Samples were loaded into 20 ml headspace vials with volumes adjusted to achieve 20 nmoles N (for DeltaV$^{PLUS}$) or 40 nmoles N (Delta$^{PLUS}$ XP). For $NO_3^-$ or $NO_x^-$ isotope samples, $\delta^{15}N$ and $\delta^{18}O$ values were calibrated using nitrate isotope

standards USGS32, USGS34 and USGS35 (Böhlke et al., 2003). Each run included two quality control samples (a GEOTRACES deep seawater sample and an in-house standard $KNO_3$ solution). Standards were run at 9-sample intervals, and used for correction of instrument drift. For $NO_2^-$ isotope samples, $\delta^{15}N$ and $\delta^{18}O$ values were calibrated using nitrite isotope standards RSIL-N23, N7373 and N10219 (Casciotti et al., 2007). These standards were run at ~6-sample intervals at two levels (5 and 10 nmol $NO_2^-$), to correct for sample size and instrument drift. The mean analytical precision of natural abundance $\delta^{15}N$ isotope measurements

using the denitrifier method is typically less than 0.5‰, but enriched experiments often have higher standard deviations. For our tracers experiments the analytical precisions were 0.4‰, 4‰ and 0.7‰ for $^{15}NO_x^-$, $^{15}NO_3^-$ and $^{15}NO_2^-$ measurements, respectively.

Filters for determination of $^{15}N$ uptake rates were dried at 50 °C overnight and packed into tin capsules prior to shipment to the Biogeochemical Stable Isotope Facility at the University of Hawaii. Samples were analyzed on a Thermo Scientific Delta V

Advantage isotope ratio mass spectrometer coupled to a Costech Instruments elemental analyzer.

### 2.5 Rate Calculations:

The rates of microbial transformations were calculated using measurements of $^{15}N$ enrichment in the product pool over time

following Eq. (1):

$$rate_t = \frac{[^{15}N]_{p,t} - [^{15}N]_{p,t0}}{F_{r,t0} \; x \; \Delta t}, \tag{1}$$

where $^{15}N_{p,t0}$ and $^{15}N_{p,t}$ are the concentrations of $^{15}N$ in the product at the start of the experiment (t0) and the final time point (t), respectively. The fraction of $^{15}N$ in the reactant N pool, $F_{r,t0} = {^{15}N}/({^{14}N}+{^{15}N})$, includes the ambient DIN and $^{15}N$ tracer addition. The detection limit was calculated as the rate that can be reasonably discerned from zero. Since variation in replicate isotope

measurements can be more variable at higher enrichment levels, we used the CV% for each rate process to help normalize across varied enrichment levels in our tracer experiments. The theoretical detection limits for each process were calculated from equation 1 by propagating a mean CV% increase in $\delta^{15}N$ into the product pool. The detection limits were 0.2, 8.5 and 0.9 nM d$^{-1}$ for ammonia oxidation, nitrite oxidation and nitrate reduction, respectively.





Uptake rates were determined using particulate samples collected at the end of each experiment. Analysis of particulate samples by isotope ratio mass spectrometry provided particulate $\delta^{15}N$ and the total particulate N (μmol N). Uptake rates were calculated following a constant uptake model as discussed in Dugdale and Wilkerson (1986). The above equation (1) was slightly modified with the assumption that the atom fraction $^{15}N$ of the ambient DIN reactant and initial particulate N are 0.003663, and the initial reactant pool was calculated from the mixture of ambient and $^{15}N$-labeled DIN based on mass balance.


Daily rates for ammonia oxidation, nitrite oxidation, nitrate reduction and nitrite uptake were calculated from hourly rates using a 12 h:12 h dark light cycle from the dark incubation and the 8 h time point from the appropriate light level, and are reported as nM $d^{-1}$. Net nitrification (NetNit) rates were calculated by subtracting nitrite oxidation rate from ammonia oxidation rate. Likewise, for phytoplankton processes net nitrite production (NetPhy) was calculated as nitrate reduction minus nitrite uptake. Furthermore,
total net nitrite production rate ($NetNO_2^-$) from all four nitrite cycling processes was calculated by subtracting consumption processes (nitrite oxidation and nitrite uptake) from the sum of the production processes (ammonia oxidation and nitrate reduction). Note that the summation of rates into a net rate will be influenced by potential $^{15}N$ enhancement occurring in each process.

**2.6 Light inhibition and enhancement**

To compare the influence of light on nitrite cycling across different source waters, a percent change in rate (R) due to light level was calculated for each experiment, relative to dark conditions ($R_{DK}$). Percent change was calculated as a fraction relative to the dark incubation and multiplied by 100 ($P_C = 100*(R-R_{DK})/R_{DK}$). Calculating percent change relative to the dark rates means that rates from the low light treatments, which approximate the *in situ* conditions at ~1% PAR, show whether the populations are
inhibited or enhanced by light in their natural environment (at collection depth). Rates showing negative percent change are considered inhibited by light, while positive percent change values (typically phytoplankton-driven processes) were enhanced by increasing light level.

**3 Results**

**3.1 Nutrients and hydrography**

The coastal station (PS3) from April 2018 (SR1805) was located 12 miles from the coast with a shallow mixed layer of ~16 m (Fig. 1). The depth of 1% PAR was at 31 m, and the nitracline fell within the euphotic zone. At the top of the nitracline (~10 m), light was ~13.6% surface PAR. With both nitrate and light available, maximal chlorophyll concentrations reached as high as 12.3 mg $m^{-3}$ at a depth of 13 m. The PNM was at 20-30 m depth, with maximum concentrations reaching 1.32 μM. A large secondary
nitrite maximum (max 2800 nM $NO_2^-$) was also detectable below ~55 m, within the oxygen deficient zone. In contrast, the offshore station (PS1) had a deeper mixed layer (~45 m) with a nitracline beginning at 50 m. Light reached deeper into the water column, with 1% PAR at ~59 m. At the offshore station, the light level at the nitracline depth was ~3.3% surface PAR. Chlorophyll levels were lower, and reached a maximum of only 6.4 mg $m^{-3}$ at a depth of 49 m. The PNM was at 55-60 m, with a concentration as high as 1.52 μM. A secondary nitrite maximum was also detectable around 220 m at the offshore station, but a much lower nitrite accumulation was found compared to PS3 (<100 nM). The central station (PS2) had a PNM nitrite concentration reaching 620 nM
situated near 65 m. The secondary nitrite maximum was large, reaching 2200 nM near 180 m. A well-defined nitracline began at 55 m, similar to the offshore station. The PS2 chlorophyll maximum reached 5.7 mg $m^{-3}$ at a depth of 64 m (Fig. 1, S1, Table S1).





During the FK180624 cruise in June 2018, Stations 2 and 9 were visited before and after a storm passed through the area, respectively. At Station 2, the primary nitrite maximum had a concentration of 766 nM at 66 m depth (SigmaT = 23.899 kg m$^{-3}$). Nitrate concentration at the depth of the nitrite maximum was 8.4 µM. At Station 9, the primary nitrite maximum had a concentration of 390 nM at 68 m (SigmaT=23.13 kg m$^{-3}$) and the nitrate concentration at this depth was 4.3 µM (Fig. 1, S1, Table S1).

**Figure 1. Map of study region showing cruise tracks from April 2018 (SR1805) and June 2018 (FK180624). Stars indicate stations where water was collected for experimental manipulations near the depth of the primary nitrite maximum.**

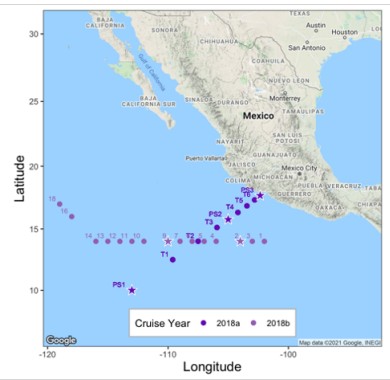

**3.2 Experimental rate measurements**

Rates for light and nitrate experiments conducted at the three processes stations (coastal PS3, central PS2, offshore PS1) during the April 2018 cruise are presented below. Experimental rates from additional stations in April 2018 and the June 2018 cruise are presented in the supplement (Fig. S2, S3, S4).


**3.2.1 Coastal station (PS3)**

At the coastal station PS3, ammonia oxidation and nitrite oxidation rates ranged from 64±0.8 to 96±1.3 nM d$^{-1}$ across all experimental treatments (Fig. 2a, b). Source water for this experiment was collected at 30 m depth (SigmaT= 24.75 kg m$^{-3}$) where

ambient light was ~1.4% of surface irradiance (Table S2). This ambient light level corresponded to the simulated light levels in the LL deck incubators (~1% surface PAR), while ML and HL incubators were ~4% surface PAR and ~20% surface PAR, respectively. Addition of nitrate resulted in an increase in nitrate$^{-}$ concentration from 18 µM to 37 µM.



**Figure 2. Rate measurements (nM d$^{-1}$) from experimental manipulation of light and nitrate using source water collected at coastal station PS3 (top row), central ODZ station PS2 (middle row) and offshore station PS1 (bottom row). Ammonia oxidation (a, e, i), nitrite oxidation (b, f, j), nitrate reduction (c, g, k) and nitrite uptake (d, h, l) are shown at each station with ambient nitrate concentration (solid bars) and 20 µM nitrate treatment (open bars) for each light condition (dark=DK, low light=LL, medium light=ML and high light=HL). Error bars depict standard error of replicate incubations where available.**

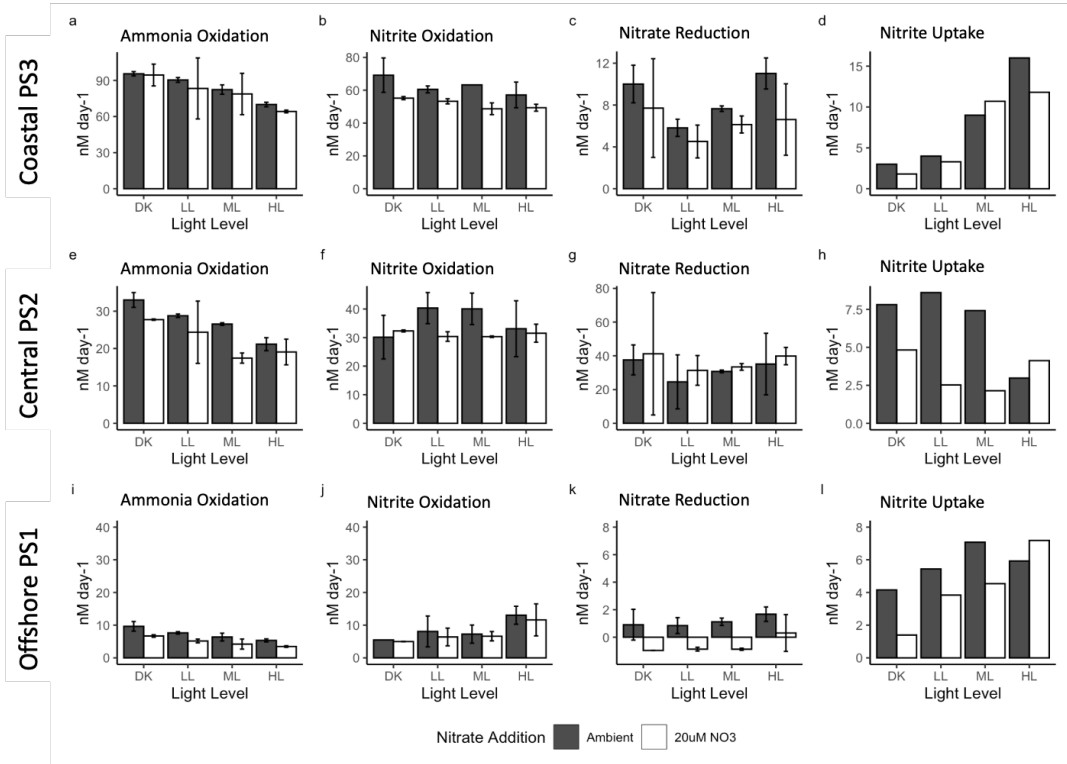


There were measurable rates of ammonia oxidation in all treatments, with the highest rates found in the dark (96±1.3 nM d$^{-1}$) (Fig. 2a). Rates of ammonia oxidation decreased as the light level increased from dark to the high light treatment, but even in the HL treatment, ammonia oxidation was still high (70±1.3 nM d$^{-1}$). This trend occurred in both the 20 µM NO$_3^-$ treatment and the ambient NO$_3^-$ treatment. In fact, duplicate experimental bottles were not statistically different between ambient and 20 µM NO$_3^-$ treatments

at PS3, except in the HL treatment (t-test p value = 0.02). Nitrite oxidation rates also declined with increasing light at PS3, but the decrease was smaller than that of ammonia oxidation. The highest nitrite oxidation rate was 69±7.4 nM d$^{-1}$ in the DK treatment and the lowest rate was 57±5.5 nM d$^{-1}$ in the HL treatment. Average nitrite oxidation rates in the 20 µM NO$_3^-$ treatments were lower than the ambient treatments for each light condition, but were not statistically different in the DK and LL conditions.

Nitrate reduction rates at PS3 ranged from 4.5±1.1 to 11±1 nM d$^{-1}$, which were much lower than the nitrification rates. However, there was not a unidirectional change across light levels, as is expected for phytoplankton-driven processes. The lowest nitrate reduction rate was in the LL treatment, and rates increased in the ML and HL conditions (7.6±0.2 nM d$^{-1}$ and 11±1 nM d$^{-1}$, respectively) resulting in a positive correlation across those light levels. However, high rates were also seen in the dark incubations (7.7±3.3 to 10±1.3 nM d$^{-1}$), which were of similar magnitude to those in the HL condition. Trends across light levels in the 20 µM

NO$_3$ treatments were similar to the ambient treatment, although rates appear to be slightly lower overall. Nitrite uptake rates ranged



from 2 to 16 nM d$^{-1}$ and were similar in magnitude to nitrate reduction rates. The nitrite uptake rates increased steadily from the dark treatment to the highest light treatment. Typically, phytoplankton take up nitrite after, or simultaneously with, nitrate so low nitrite uptake in the presence of 18 or 37 μM nitrate is not surprising. Further, nitrite uptake in the 20 μM NO$_3^-$ treatments were generally lower than the ambient NO$_3^-$, but as there are no replicates for nitrite uptake determining statistical significance is not possible.

**Figure 3. Rates of (a) ammonium and (b) nitrate uptake from experimental manipulation of light and nitrate from source water collected at Station PS3. Ambient nitrate concentration (solid bars) and 20 μM nitrate treatment (open bars). No replicates are available for these experiments (n=1).**

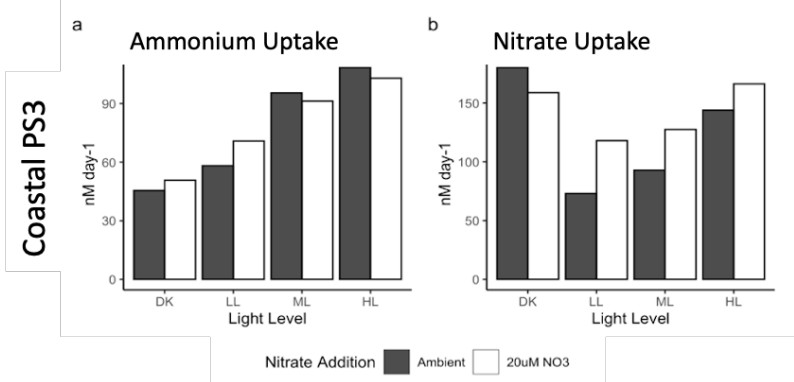

At station PS3, additional measurements of DIN uptake were collected. Ammonium uptake rates were on the same order of magnitude as the ammonia oxidation rates, ranging from 45 to 108 nM d$^{-1}$ across the light conditions (Fig. 3a). However, unlike ammonia oxidation, ammonium uptake was positively correlated with light level, with the highest rates observed in the HL condition. Addition of 20 μM NO$_3$ resulted in slightly higher ammonia uptake rates in the DK and LL treatments, although there were no replicates. Ambient ammonium concentrations were low, and addition of 200 nM $^{15}$N-NH$_4^+$ tracer may have enhanced the ammonium uptake activity. Ammonium uptake rates were 6-20x higher than nitrite uptake.

Nitrate uptake rates were the highest of any measured rates, ranging from 73 to 180 nM d$^{-1}$. The lowest rates were found in the LL treatment and increased in the ML and HL treatments (93 nM d$^{-1}$ and 144 nM d$^{-1}$, respectively) (Fig. 3b). As observed in the nitrate reduction measurements, high nitrate uptake rates were also observed in the DK incubation and were on par with the HL treatment. In each light treatment (but not in the dark incubation), the 20 μM NO$_3^-$ treatment led to an increase in nitrate uptake rate, although lack of replication limits determination of statistical significance.

### 3.2.2 Central station (PS2)

The nitrification rates at station PS2 were more moderate than at station PS3 (Fig. 2e, f). Water for these experiments was collected from 75 m depth (SigmaT= 25.04 kg m$^{-3}$), just below the nitrite maximum at PS2 where light was ~2% of surface irradiance (Table S2). Ambient nitrate concentration in the source water was 16 μM prior to experimental nitrate addition.

Ammonia oxidation rates ranged from 17±1 to 33±1.4 nM d$^{-1}$ across experimental treatments at station PS2 (Fig. 2e), with the highest rates in the DK incubation and the lowest rates in the HL condition. Ammonia oxidation rates appeared to be reduced in





the 20 μM NO$_3^-$ treatments, especially in the ML condition, where the ambient nitrate treatment had a rate that was 1.5x higher than that with nitrate addition. Nitrite oxidation rates ranged from 30±0.2 to 40±3.8 nM d$^{-1}$ across all treatments. There was no uniform directional response of nitrite oxidation rates to increases in light level, but it is notable that the rates did not strongly decrease with increased light. The rates in ambient NO$_3^-$ treatments were ~30 nM d$^{-1}$ in both the DK and HL treatments, while the LL and ML treatments had ambient rates near 40 nM d$^{-1}$. The 20 μM NO$_3^-$ treatments all had measured nitrite oxidation rates near ~30 nM d$^{-1}$ regardless of light level (Fig. 2f).

Nitrate reduction rates were 24.6±11.3 to 41.2±25.7 nM d$^{-1}$ across treatments at PS2 (Fig. 2g). Similar to station PS3, the lowest rates were in the LL treatment. The DK and HL treatments had the highest rates (near 40 nM d$^{-1}$), while the LL and ML treatments had lower rates. The addition of 20 μM NO$_3^-$ did not appear to clearly change the rates of nitrate reduction at any light level. Nitrite uptake rates ranged from 2.1 to 8.6 nM d$^{-1}$ across treatments and there was no unidirectional response with increasing light level (Fig. 2h). The lowest rates were seen in the LL and ML treatments with the addition of 20 μM NO$_3^-$. The highest rates were also seen in the LL and ML treatments but in the ambient NO$_3^-$ treatment. No additional nitrogen uptake rates were analyzed at station PS2.

### 3.2.3 Offshore station (PS1)

Rates of N transformation at station PS1 were generally lower than at PS2 and PS3 (Fig. 2i-l). Water for these experiments was collected at 60 m depth (Sigma T= 23.82 kg m$^{-3}$), just below the PNM feature at light levels near 0.5% of surface PAR. At 60 m depth, the ambient nitrate concentration was ~12 μM, so the NO$_3^-$ addition treatment had 32 μM (Table S2).

Ammonia oxidation rates ranged from 3.5±0.2 to 9.7±1 nM d$^{-1}$, with the highest rates seen in the dark treatments. These rates followed the same light response pattern seen at the coastal station, with the highest rates in the DK, decreasing into the HL treatment. The addition of 20 μM NO$_3^-$ to the incubations slightly decreased ammonia oxidation rates in every light treatment, although this trend was not always statistically significant. The nitrite oxidation rates at the offshore station were much lower than those measured at stations PS2 and PS3. The range in rates across treatments was 5±0.1 to 13±1.9 nM d$^{-1}$, with the highest rates occurring in the HL condition (~13 nM d$^{-1}$). In contrast to the ammonia oxidation rates, the offshore nitrite oxidation rates increased as light increased in both the ambient and 20 μM O$_3^-$ treatments.

The nitrate reduction rates at station PS1 were very low, with all rates lower than 2 nM d$^{-1}$. The highest rate was in the HL treatment, and rates decreased to below the detection limit in many of the 20 μM NO$_3^-$ treatments. While nitrate reduction was minimal, nitrite uptake was still active at the offshore station, with ambient rates ranging from 4.2 to 7.1 nM d$^{-1}$. There was an increase in nitrite uptake with increasing light. Nitrate additions may have decreased nitrite uptake rates, especially in the lower light treatments. No additional nitrogen uptake rates were analyzed for station PS1.

### 3.3 Light effects on the balance of nitrite production and consumption processes

Ammonia oxidation dominated nitrite production in the LL treatment (which is comparable to the ambient light at source water collection depth) across the three stations (Fig. 4a,b,c). At the coastal and offshore stations, ammonia oxidation comprised more than 85% of nitrite production in LL, while the percentage was smaller at the central station (~54%). Increased light tended to





increase the relative proportion of nitrite derived from nitrate reduction. The increase in the relative contribution of nitrate reduction to nitrite production was driven by both increased nitrate reduction rates and decreased ammonia oxidation rates (Fig. 2).

**Figure 4. Relative contribution of production processes (top row) and consumption processes (bottom row) across light treatments at coastal PS3 (a,d), central PS2 (b,e) and offshore PS1 (c,f). Nitrification processes are black and phytoplankton processes are white. Ambient nitrate treatments only, does not include 20 μM NO$_3^-$ treatments.**

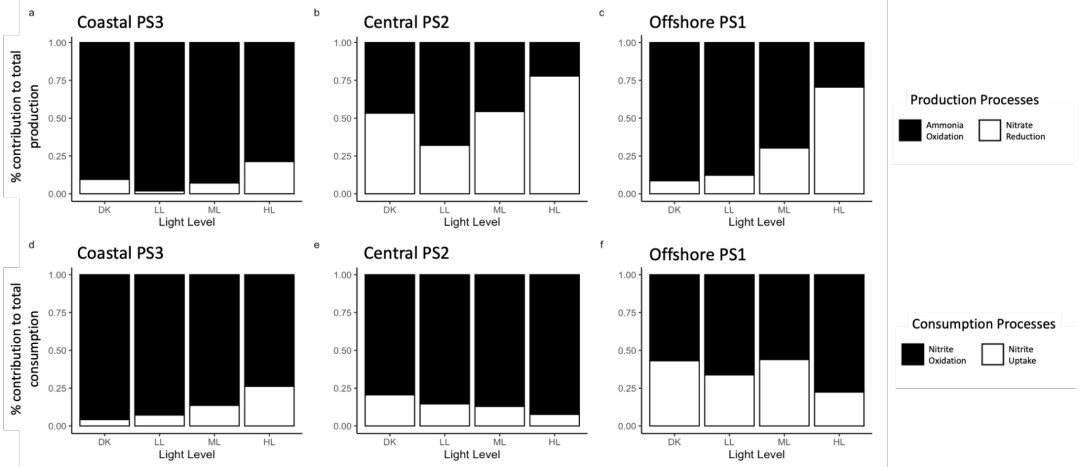

Nitrite oxidation dominated nitrite consumption at all stations (Fig. 4d,e,f). In LL treatments at the coastal and central stations, nitrite oxidation was responsible for over 80% of nitrite consumption, while nitrite oxidation comprised approximately 60% of nitrite consumption at the offshore station. Increasing light did not exert a uniform directional influence: the proportion of nitrite consumed by nitrite oxidation declined at the coastal station but increased at the central and offshore stations. While nitrite oxidation comprised a larger percentage of overall consumption at the offshore station, the rates were the lowest observed (Fig. 2j).

### 3.4 Percent change in rates due to light treatments

In general, ammonia oxidation rates were inhibited by increased light while phytoplankton activity was enhanced. The largest percent change ($P_c$) of ammonia oxidation was seen in the HL condition at the offshore station, which reached 45% (Fig. S5i). At the coastal station, the HL condition reduced ammonia oxidation rates by 27%. Low light conditions (which correspond most closely with light level at collection depth) showed that ammonia oxidation rates within this source water microbial community are already 5-20% inhibited in the field, with the highest ambient light inhibition observed at the offshore collection depth (Fig. S5i). Nitrite oxidation rates were expected to show light inhibition as well, but only the coastal station showed clear inhibition, where $P_c$ reached 17% in the HL treatment (Fig. S5b). At all stations, the response of nitrite oxidation to increasing light levels was not as consistent as the responses seen in ammonia oxidation rates (Fig. S5). Surprisingly, nitrite oxidation rates at the offshore station increased by >25% at all light levels relative to the dark in both ambient and 20 μM NO$_3^-$ treatments. However, the magnitude of those rates was quite small (Fig. S5j, Fig 2j). Nitrate reduction rates were enhanced with increasing light beyond LL, but were also enhanced in the dark treatments at PS3 and PS2. Percent change in nitrite uptake was much larger than changes seen



in nitrification rates (Fig. S5 d,h,i). The coastal station nitrite uptake rates had the largest response to increased light, with rates increasing 300-650% in the HL treatments, relative to the dark.

The $P_c$ calculations for each station (including all cruises and $NO_3^-$ treatments) can be summarized to look for general patterns that may hold across the region. Ammonia oxidation showed a consistent light effect across stations, where rates declined progressively

in the higher light treatments. The summary plot reflects this pattern with declining percent change in rate as light level increases (Fig. 5a) with a small range in values falling within the 0.25-0.75 quantiles. The mean percent change in nitrite oxidation showed increasing rates with increasing light level across the region (Fig. 5b). However, nitrite oxidation had varying directional response to increasing light across stations (Fig. 2b,j,f, S2b, S3b, S4b,d), which contributes to the large range in $P_c$ at LL and is obscured in the averaging of stations (Fig. S5 c,g,k).


**Figure 5. Summary plots of percent inhibition for each process across stations and depth by light treatment. (a) ammonia oxidation, (b) nitrite oxidation, (c) nitrate reduction, and (d) nitrite uptake at each experimental light level (LL = low light, ML = medium light, HL = high light). The horizontal bar is the mean % inhibition, the box depicts 0.25-0.75 quantiles, and lines show range to 0.95 and 0.05. Treatment different from DK with t-test, ** p<0.01, * p<0.1.**

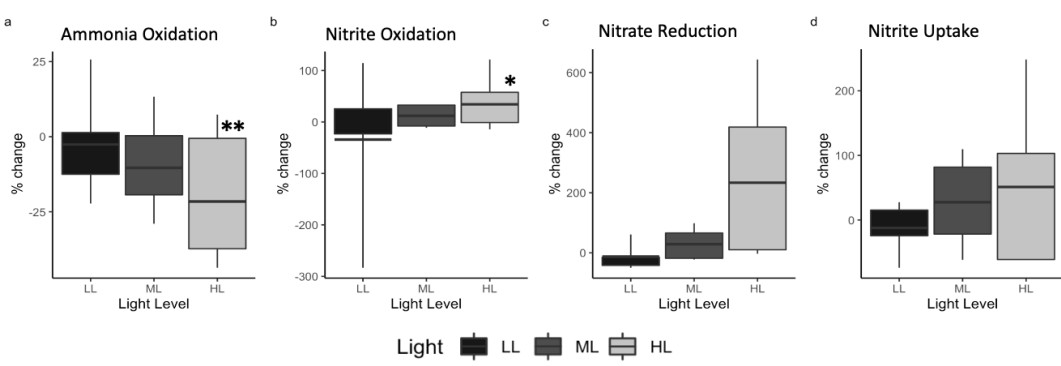


The percent change in nitrate reduction rates increased with light level, but there was high variation in the station data for this process (Fig. 5c). The range in data within the 0.25 to 0.75 quantiles is larger than for nitrification rates (~4x) especially in the HL treatments. Percent change in nitrate reduction exhibited some positive responses to decreased light (e.g., coastal station and central

station) and discrepancy between the ambient and 20 µM $NO_3^-$ treatments (e.g., offshore and central) that contributed to the wide range of data falling within the 0.25 to 0.75 quantiles. Summarized percent change in nitrite uptake also had a wide range in the 0.25 to 0.75 quantiles, as the directional response to light was not consistent across the coastal, central, and offshore stations (nitrite uptake at the central station declined with increased light Fig. 2h). Generally, the percent change in nitrite uptake showed enhancement of rates with increased light level, and was likely driven by the strong light response seen at the coastal station (PS3;

Fig. 2d). The variance in nitrate reduction and nitrite uptake rates across stations was higher than that for nitrification rates.

### 3.5 Net nitrite production under varying light levels

Net nitrite production from nitrification (NetNit, ammonia oxidation minus nitrite oxidation) declined with increasing light at each

station (Fig. S6), consistent with field observations that NetNit generally decreases from the base of the euphotic zone towards the surface (Travis et al., 2023). The dark treatments had the largest net positive rate of nitrite production from nitrification at all





stations. Coastal station PS3, which had the largest rates of both ammonia oxidation and nitrite oxidation, also had the largest NetNit production rates with positive rates in every light treatment (Fig. S6a,d). At some stations, light level modulated NetNit enough to flip net production rates from positive to negative (Fig. S6d). Both central and offshore stations had negative NetNit

values in all light treatments except DK, and had smaller contributing rate magnitudes (Fig. S6b,c). Patterns were similar in the 20 μM $NO_3^-$ treatment (Fig. S7)

Net nitrite production from phytoplankton (NetPhy, nitrate reduction minus nitrite uptake) declined with light at the offshore and coastal stations, and the offshore station had negative NetPhy rates at all light treatments (Fig. S8). While nitrate reduction

increased with light at the coastal station, it was not large enough to offset the corresponding increase in nitrite uptake (Fig. S8a). The large positive NetPhy values at the central station were driven by very high nitrate reduction rates (Fig. S8b). These patterns held in the 20 μM $NO_3^-$ treatment (Fig. S9)

Overall net nitrite production rates (NetNO$_2$) were calculated by combining all four measured nitrite cycling processes (Fig. 6).

There were differences in magnitude and sign for NetNO$_2$ across the coastal, central, and offshore stations, but NetNO$_2$ generally declined with increasing light level in both the ambient and 20 μM $NO_3^-$ treatments. The offshore station had negative NetNO$_2$ rates at all light levels and $NO_3^-$ treatments except for the ambient $NO_3^-$ dark treatment (Fig. 6c) which had a small positive rate (<2 nM day$^{-1}$). The coastal and central stations had positive NetNO$_2$ across all light and nitrate treatments (Fig. 6a,b). The 20 μM $NO_3^-$ treatments showed similar NetNO$_2^-$ patterns to the ambient treatment, except for in the LL condition.


**Figure 6.** Component processes of net nitrite production (NetNO$_2$) at each station: coastal PS3 (a) central PS2 (b) and offshore PS1 (c). Positive values represent production of nitrite and negative values represent consumption of nitrite. The net nitrite production rates at each station are presented together in panel d, and overlaid as white dots on individual station panels a,b,c. The 20 uM NO3 treatments are on the left in panels a,b,c, and as dashed lines in panel d. Errors are omitted, but presented in Figure 2 for individual rates. Pooled

error is calculated for NetNO$_2$ (d).

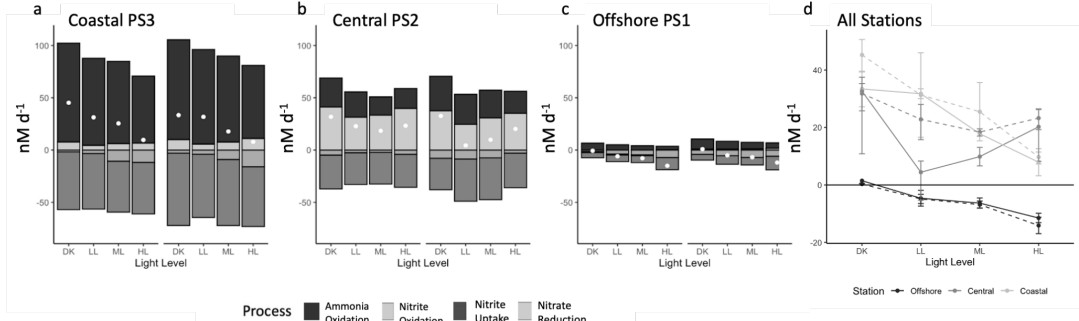

## 4 Discussion:

### 4.1 Sensitivity of nitrification to light

Light inhibition has been used as a mechanism to help explain exclusion of ammonia-oxidizing and nitrite-oxidizing microbes from the surface ocean (Lomas and Lipschultz, 2006; Olson, 1981a). However, active nitrification has been observed in the sunlit ocean (Beman et al., 2008; Clark et al., 2008; Santoro et al., 2013; Smith et al., 2014; Francis et al., 2005; Ward, 2005) and there



is variation in apparent photosensitivity across natural communities of ammonia oxidizers (Qin et al., 2014). While decreased ambient nitrification rates have been seen above the nitrite maximum in previous work (Travis et al., 2023; Beman et al., 2012,
2013), those studies were not able to conclude that lower rates were caused by increased light level, because the microbial community and cell abundance were different at each depth measured in a vertical profile. Since the experimental design in the current study manipulated light using a single source water community, we can conclude that the declines in bulk rate measurements observed here were most likely caused directly or indirectly by changes in light intensity. Ammonia oxidation occurred in all our light treatments, including the highest level (~20% PAR), showing that light did not completely inhibit ammonia
oxidation in our samples (Fig. 2).

In the field, the influence of light is overlaid on top of other factors that control the distribution of microbial populations in the natural environment, where environmental pre-conditioning and cell abundance may set the variance in the baseline rates measured in these experiments. While the magnitude of the ambient ammonia oxidation rates were indeed very different between stations
(e.g., 91 and 8 nM day$^{-1}$ at coastal PS3 and offshore PS1, respectively), the percent light inhibition of ammonia oxidation rates when moved from ambient (~1% surface PAR) to high light treatment were similar (27-45% at all three stations).

The source water collected at each station had ambient PAR levels of 0.5~2% of surface irradiance, which were most closely approximated by the low light incubation tank (~1% PAR). Comparison of ammonia oxidation rates between the LL treatment
(1% surface PAR) and the corresponding dark incubations indicates that ammonia oxidation was inhibited by 5-21% at *in situ* conditions (Fig. S5). This is consistent with prior results from the North Pacific Ocean, which showed there was 25-41% inhibition of ammonia oxidation rates at the depth of 1% surface PAR (Horak et al., 2018). This is in contrast to Smith et al. (2014) who saw little inhibition of ammonia oxidation when water from near the PNM was incubated at 50% surface PAR. Ammonia oxidation rates at the offshore station PS1 were low (8 nM d$^{-1}$), already light inhibited (by 21%), and more sensitive to increased light
compared to the coastal community (45 vs 26% inhibited in HL, respectively) (Fig. 2i, Fig. S5). This could result from offshore source water communities being acclimated to more stable, low light conditions compared to cells in dynamically changing light fields closer to the coast.

Nitrite oxidation rates were similar in magnitude to ammonia oxidation rates at each station, but an overarching light response for
nitrite oxidation was not easy to discern. Olson et al. (1981a) showed that nitrite oxidation rates in field studies were inhibited by increases in light, possibly even more than ammonia oxidation. However, other work has suggested that nitrite oxidation may be less sensitive to increasing light intensity compared to ammonia oxidation, although recovery after photoinhibition is slower (Horak et al., 2013; Guerrero and Jones, 1996b). Oceanic profiles of nitrite oxidation activity show vertical distributions that are shaped similarly to ammonia oxidation, with maximal rates at the base of the euphotic zone and lower rates in surface waters (Travis et
al., 2023; Beman et al., 2012). Overall, our data suggest that light is not the primary reason for generally low rates of nitrite oxidation in the surface ocean, but may still play a role under some conditions, in addition to substrate supply and competition for substrate within the community.

We targeted source water from PNM depths where ammonia and nitrite oxidizers are typically abundant and active (Santoro et al.,
2010, 2013; Travis et al., 2023). Challenging these communities with higher light levels simulated water moving upward in the water column. In some HL treatments we measured rates of ammonia oxidation higher than those typically found in shallow, high-





light environments where ambient nitrifier abundance is low (Santoro et al., 2010; Beman et al., 2013, 2012). Figure 7 shows the measured rates from the experimental light manipulations on top of ambient measurements of ammonia oxidation and nitrite oxidation collected from a variety of depths and light conditions in the ETNP. This comparison illustrates that while nitrification

rates are inhibited by light to an extent, the HL rates are much higher than ambient rates measured in communities collected from comparable light levels in the field. We argue that this is partly due to the lower abundance of nitrifying organisms in shallow waters due to chronic substrate limitation and competition. This idea is illustrated quantitatively by modeling (Zakem et al., 2018) that shows substrate concentrations (resource limitation) can control the exclusion of ammonia- and nitrite oxidizers from surface waters without invoking light inhibition.


**Figure 7. Experimental rates from light experiments (open circles) plotted on top of ambient measurements (closed circles) from the ETNP region. The % surface irradiance are from light levels in experimental incubations tanks, or % surface irradiance from collection depth for ambient rates. Measurements from the same source water are connected with a line across the experimental light levels tested.**

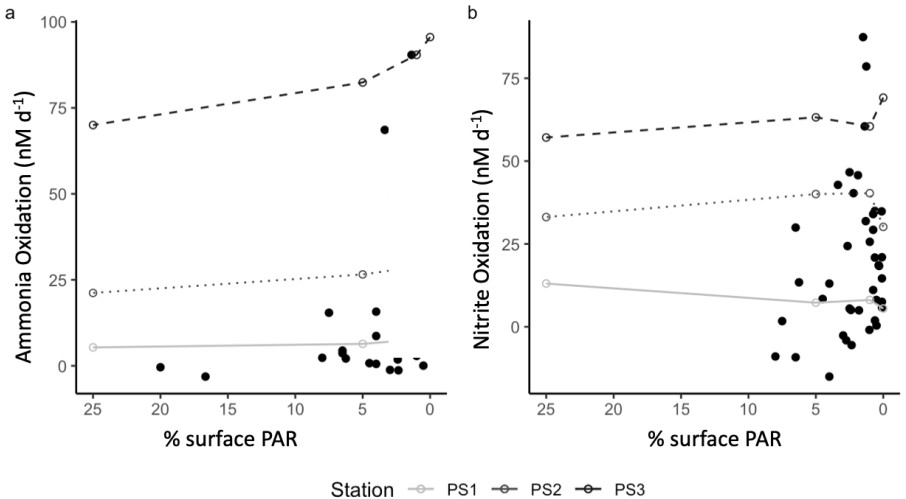


While it appears from our data that light inhibition alone does not exclude nitrification activity from the surface ocean, differential light responses could still help shape the balance of the two steps of nitrification vertically across the PNM feature. Decoupling of the two steps of nitrification is commonly observed in the field (Heiss and Fulweiler, 2016), whether caused by changes in mixing

(Haas et al., 2021), temperature (Schaefer and Hollibaugh, 2017) or light (Olson, 1981a), and is often invoked to explain nitrite accumulation in the surface ocean. Observations of ammonia oxidation rates reaching maxima slightly shallower than nitrite oxidation maxima were initially interpreted as differential light tolerance of the two steps of nitrification in the surface ocean (Olson, 1981b). However, this vertical structure is less clear in larger datasets of paired ammonia oxidation and nitrite oxidation rates (Beman et al., 2013; Travis et al., 2023), and observational patterns cannot be causally attributed to different light tolerances.


Balanced nitrification may be more likely when rates of individual steps are low (e.g., PS1, Fig. 6c). In the ETNP, net nitrification (NetNit) rates have the largest imbalance around the depths of highest nitrifier activity, at the base of the euphotic zone near the PNM, but a NetNit maximum is not guaranteed at this depth (Travis et al., 2023). Moving natural microbial communities from



near the PNM abruptly into high light levels caused a decline in NetNit that was driven by a larger decline in ammonia oxidation

rate compared to nitrite oxidation. For example, there was a 20 nM d$^{-1}$ difference in ammonia oxidation rate between LL and HL treatments at the coastal station PS3, and a corresponding decline of only 3 nM d$^{-1}$ in nitrite oxidation (Fig. 2). Ammonia oxidation appears to be more consistently inhibited in our light experiments compared to a more variable response in nitrite oxidation, although NetNit shows a clear decrease in rate with increasing light (Fig. 2, Fig. 6). If this pattern of net consumption of nitrite at higher light levels holds more generally, it could potentially help to define the upper boundary of nitrite accumulation in the PNM

(Fig. 6d).

Previous work has suggested that ammonia oxidizers recover more quickly from light inhibition than nitrite oxidizers, and this differential recovery could also cause nitrite to accumulate at the PNM. This suggestion came from experiments done on lab cultures of ammonia-oxidizing bacteria, which we now know are typically less abundant than ammonia-oxidizing archaea in the

surface ocean and contribute less to total ammonia oxidation activity (Beman et al., 2012). Additionally, *Nitrococcus maritimus* and *Nitrobacter sp.* were used in the light recovery experiments (Guerrero and Jones, 1996b), but we now know *Nitrospina sp.* are more abundant and active in this region. More recent work using ammonia oxidizing archaea species shows that archaea may be even more sensitive to light than bacterial ammonia oxidizers (Merbt et al., 2012).

Recovery from light inhibition can be seen in DK treatments, where ammonia oxidation rates increased when shifted from LL into the dark, while the directional response of nitrite oxidation rates was more variable. Moving the nitrite oxidizing community from low light to dark conditions only led to increased nitrite oxidation (i.e., recovery from light inhibition) at coastal station PS3, not at central station PS2 or offshore station PS1. Since ammonia oxidation appears to drive the patterns in NetNit from the nitrifier community, the focus of most studies on ammonia oxidation rate measurements may be justified. However, rates of nitrite oxidation

are of similar magnitude to ammonia oxidation rates, which indicates that they are important for overall nitrite cycling in the surface ocean. As evidenced by the ubiquity of nitrite oxidoreductase enzyme (Nxr), more studies on the activity and environmental response and high variability of nitrite oxidizer communities are needed (Saito et al., 2020).

**4.2 Light sensitivity of phytoplankton nitrite release and nitrite uptake**


We hypothesized that nitrate reduction would increase with increasing light, since the source water in all experiments was nitrate replete (>12 $\mu$M). In general, the rates of phytoplankton-driven processes were indeed enhanced by increases in light. Nitrate reduction rates increased slightly with light intensity across stations, but the magnitude of nitrate reduction was still much smaller than that of ammonia oxidation at the coastal and offshore stations. These incubations were done over 8 h, which is likely too short

to capture nitrate reduction enhancement due to an increase in nitrate reductase enzymes, as the enzymes are synthesized over a daily cycle (Berges, 1997). The positive light response we observed likely reflects the enhanced activity of pre-existing nitrate reductase, and not de novo synthesis or cell growth. Therefore light response patterns in nitrate reduction will be constrained by the initial characteristics of the source community at each station. Nitrate reduction in this dataset was measured as an increase in nitrite released from the cell, not as assimilation into particulate matter, allowing a unique perspective on nitrite dynamics. Previous

work has shown that nitrite release by phytoplankton is linked to nitrate uptake rates, with ~10% of nitrate uptake potentially being released as nitrite on average, although some estimates from N-limited or N-starved cultures had release rates upwards of 25% (Collos, 1998; Kiefer et al., 1976; Martinez, 1991). Enhanced nitrite release has been found in cells that were recently nitrogen





limited, and nitrite release appears to be a transient response to cells that are adjusting their growth rates and nitrogen assimilation enzymes to accommodate a resupply of nitrogen (Sciandra and Amara, 1994). Based on nitrate uptake measurements at the coastal

station, our nitrite release rates (nitrate reduction rates) were 4-8% percent of nitrate uptake rates (Fig. S8). High rates of nitrite release >20 nM d-1 (nitrate reduction measurements) were also observed at coastal stations in the ETNP in a prior study (Travis et al., 2023). For dynamic water columns, such as near coastal upwelling regions, changes in light and nitrate supply may induce more frequent episodes of nitrite release as cells frequently re-acclimate to new conditions.

Nitrite release from phytoplankton has been suggested as a consequence of energy balancing, where sudden increases in photon flux are dissipated by nitrate reduction while nitrite reduction is rate limiting. Diatoms may use nitrate reduction to avoid photodamage during high light events, resulting in release of nitrite or other dissolved organic nitrogen forms (Lomas et al., 1999; Lomas and Glibert, 1999). Temporary release of nitrite may occur under more moderate conditions too, when growth limitations are alleviated (light or iron), leading to a transient mismatch between energy supply and nitrogen assimilation capabilities of the

phytoplankton community (Milligan and Harrison, 2000; Sciandra and Amara, 1994).

Interestingly, increased rates of nitrate reduction were also seen in the dark incubations (coastal station PS3 and central station PS2) where we would not expect phytoplankton to require photoprotective mechanisms or have excess energy supply. The elevated nitrate reduction rates seen in both the DK and HL conditions (compared to LL) may reflect the activity of two separate

physiological mechanisms controlling nitrite release. While nitrate and nitrite reduction are both typically light dependent processes, depending on the enzymatic catalysis of N substrate with 2 or 6 electrons respectively, many algae are capable of nitrate uptake and assimilation in the dark without active photosynthesis. Diatoms are known to continue high rates of nitrate assimilation in the dark after daytime access to high light conditions (Clark et al., 2002).

Nitrate reduction is often the rate limiting step in nitrate assimilation, as evidenced by the accumulation of nitrate within phytoplankton cells compared to nitrite rarely accumulating (Dortch et al., 1984) and that nitrite reductase enzyme (NiR) activity is typically higher than nitrate reductase enzyme (NR) in nutrient replete cells (Milligan and Harrison, 2000). Milligan and Harrison (2000) witnessed a decline in NiR enzyme activity and an increase in nitrite efflux from diatoms during conditions when photosynthetically produced reductant was low, suggesting that nitrite reduction can become the rate limiting step under reduced

reductant availability. This situation could occur chronically during low light or in the dark. Nitrite release due to incomplete nitrate assimilation during periods of light-limitation has also been observed in diatoms by Vacarro and Ryther (1960). Further investigation into the mechanisms and transient conditions for nitrite release in the dark are needed.

Nitrite uptake was also observed to be light-dependent, with nitrite uptake rates generally increasing with light (Fig. 2). Ambient

nitrite uptake rates were similar in magnitude between coastal and offshore stations, but the HL treatment appeared to enhance nitrite uptake more at the coastal station (3-fold vs 2-fold). This change in bulk rate may be partially explained by higher chlorophyll concentrations in the coastal station source water. At the coastal station, nitrite uptake in the dark treatment did not increase alongside the increased nitrate reduction, suggesting that nitrite uptake is regulated separately from nitrate uptake/reduction and does not simply "ride-along" with nitrate uptake.




Phytoplankton cannot be generalized as simply net consumers or net producers of nitrite across stations and light levels, as NetPhy (nitrate reduction minus nitrite uptake) varied from positive (net producing) to negative (net consuming) between stations and between light treatments at a single station (Fig. S8, S9). When NetPhy declined clearly with increasing light, it was driven by the strong increase in nitrite uptake rates (Fig. S8a). Interestingly, nitrate reduction rates (nitrite release) were much higher at station

PS2 than the other two stations, resulting in positive NetPhy across all light treatments (Fig. S8b, S9b). However, chlorophyll was not significantly more abundant at PS2; in fact, PS3 had the largest chlorophyll maximum. It is possible, instead, that the phytoplankton community at PS2 had a different species composition than the other stations causing larger nitrite release rates. The lack of stability in the water column chlorophyll profiles collected over the repeated casts from station PS2 is also suggestive of fluctuating conditions at this station that could cause physiological responses that stimulate nitrite release (Fig S1).


### 4.3 Net community nitrite production in response to light

At all stations there is a general unidirectional light response, where community $NetNO_2$ declines with increasing light (Fig. 6d). This correlates spatially with the upper slope of the PNM, where nitrite concentrations in the ETNP decline precipitously moving

upward toward the surface. The coastal station samples consistently showed net positive $NetNO_2$ across the light treatments and the offshore station showed mostly net consumption of nitrite; however, we observed higher nitrite accumulation at the offshore station (800 nM vs 470 nM) (Fig. 8 and Table S2). This discrepancy may be due to temporal mismatch between an instantaneous rate measurement from a single day and the time-integrated accumulation of nitrite observed at the PNM.

In a prior study, production of nitrite was dominated by ammonia oxidation below the PNM and shifted to a higher contribution from phytoplankton above the PNM (Travis et al., 2023). The light experiments show how an increase in light can cause a relative shift in the balance between ammonia oxidation and nitrate reduction, with phytoplankton contributing a larger percentage under higher light (Fig. 4). However, there is variation between stations in whether nitrate reduction becomes the dominant production process, which likely depends on the microbial population. Consumption of nitrite around the PNM is driven by nitrite oxidation

in the ETNP (Travis et al., 2023). Here, nitrite oxidation remained the dominant nitrite consumption process across all stations and light levels, even when nitrite uptake has an increasing response to increasing light (Fig. 4). Sato et al. (2022) incubated PNM water in low light and high light conditions, and concluded from decreases in nitrite concentration that nitrite was not released by phytoplankton in the eastern Indian Ocean. However, we see that the response of individual nitrite cycling processes (such as increases in nitrate reduction/nitrite release) can be obscured when summed into a net nitrite production rate.


Source water was collected at the same relative PNM depth at both the coastal and offshore stations to allow comparison of response dynamics. However, attributes of the source water and community such as cell abundance, nitrogen and light acclimation history and substrate availability are not exactly the same at each station. Responses in our light experiments are thus overlain on variations in source water characteristics. Chlorophyll concentrations were higher at the coastal station PS3 compared to offshore station PS1

(~3.5 vs. 0.4 mg m$^{-3}$) (Table S1). Slightly higher rates of bulk nitrate reduction and nitrite uptake at the coast may be explainable by higher chlorophyll concentration, but significantly higher ammonia oxidation rates at the coastal station cannot be explained by higher ammonia oxidizer abundance (*amoA* gene abundance shown in Table S2, Frey et al., 2023). The coastal station PS3 source water was collected from a depth with ambient light level of 1.4% surface irradiance, while the offshore source community was collected from a depth with 0.5% surface irradiance. The slightly higher light acclimation level of coastal ammonia oxidizers may



explain why light inhibition was lower at the coast compared to offshore (Fig. S5). Additionally, the coastal station PS3 source
water had slightly higher nitrate and ammonium concentrations compared to offshore. Work by Xu et al. (2019) showed that light
inhibition of ammonia oxidation occurred irrespective of substrate limitation or saturation, so variation in source water ammonium
or enhancement due to ammonium tracer addition should not have interfered with the observed light response.

**4.4 Role of nitrate in stimulating or suppressing nitrite accumulation**

The accumulation of nitrite at the base of the euphotic zone has been spatially correlated with the nitracline, suggesting a
relationship between nitrate and nitrite cycling (Herbland and Voituriez, 1979; Travis et al., 2023). Of the microbial processes that
mediate nitrite accumulation in the PNM, only nitrate reduction by phytoplankton is directly dependent on nitrate as a substrate.
Nitrate is a key nitrogen source for phytoplankton growth, and its presence has been shown to influence the nitrogen physiology
and regulation of cells (Berges, 1997; Fernández et al., 2009). Nitrite release by phytoplankton has been connected to nitrate
availability and uptake, where nitrite release rates can be anywhere from 5-30% of nitrate uptake depending on light levels(Olson
et al., 1980; Collos, 1998). Wada and Hattori (1971) showed that nitrite production increases as nitrate concentration increases.
Our experimental addition of 20 μM $NO_3^-$ did not clearly enhance nitrate reduction as might be expected from these earlier studies
(Fig. 2, Olson et al., 1980; Collos, 1998), and nitrate reduction at PS3 and PS1 actually declined with nitrate addition. This lack of
rate enhancement may be explained by the *in situ* nutrient status of source phytoplankton across experiments, as source water was
collected on the underslope of the PNM, solidly in the nitracline, and initial nitrate concentrations for these experiments ranged
between 7.5-19.6 μM (Table S2). Sciandra and Amara (1994) suggested that nitrite release is a transient response that occurs when
nitrate uptake suddenly increases, either by increase in N-substrate availability or light. The nitrate-replete condition of microbes
collected from these depths may have prevented a large response to additional 20 μM $NO_3^-$, as nitrate was likely not limiting for
growth or activity. Nitrite uptake might also be expected to increase with the addition of nitrate, as nitrate uptake into many
phytoplankton cells is mediated by NRT2 transporters that can take up both $NO_3^-$ and $NO_2^-$ (Sanz-Luque et al., 2015). However,
even though nitrate uptake increased with nitrate addition (at PS3, Fig 3b), nitrite uptake was lower in most of the $NO_3^-$ addition
treatments (Fig. 2).


An indirect influence of nitrate on nitrification has been suggested, whereby nitrate-based phytoplankton growth eventually
supplies ammonium substrate via grazing and regeneration (Mackey et al., 2011), or by mediating ammonium availability through
reduced competition for DIN resources (Smith et al., 2014; Wan et al., 2018; Xu et al., 2019). However, in our 20 μM $NO_3^-$ addition
experiments, we did not observe an increase in ammonia oxidation rates at any of the SR1805 stations (Fig. 2, S4). In fact, many
ammonia oxidation rates declined slightly with nitrate addition. While nitrate uptake rates at the coastal station PS3 did increase
slightly with nitrate addition (Fig. 3b), the corresponding ammonium uptake rates declined only at higher light levels (Fig. 3a), and
no corresponding increase in ammonia oxidation was observed. It is possible that an 8 h incubation is not long enough to reflect
the cascade of adjustments required to result in increased ammonia oxidation rates.

Oligotrophic phytoplankton communities have adapted to low DIN conditions, and typically maintain low $K_s$ values and high $V_{max}$
values for ammonium uptake (MacIsaac and Dugdale, 1969, 1972). Xu et al. (2019) compared substrate kinetics of ammonia
oxidation and ammonium uptake, and confirmed that phytoplankton are more competitive for ammonium substrates at low DIN
and higher light intensities. Perhaps further work on the enzymatic responses of nitrate reductase, nitrite reductase, glutamine



synthetase and ammonia monooxygenase during nitrate intrusion would clarify the interaction between microbial nitrogen physiologies near the PNM. Overall, nitrate additions did not appear to significantly alter NetNO$_2$ across stations.

**4.5 Insights into the formation and maintenance of a PNM**

Increased light intensity modulated individual microbial processes to different extents and led to changes in net nitrite production
as well as changes in relative contributions to nitrite production and consumption. Generally, we found that increased light levels cause declines in rates of ammonia oxidation and increases in phytoplankton activity. The experimental design used in this study, with discrete changes in light condition, most closely mimics dynamic conditions in coastal waters or dramatic mixing events such as storms. Net nitrite production at the offshore station is unlikely to be controlled by dramatic changes in light field, as the community is acclimated to more stable conditions, and is less likely to experience disturbances. We saw the strongest pre-existing
light inhibition of nitrification rates from this offshore community, as well as the strongest response to increases in light (Fig. S5). While ammonia oxidation rates were also inhibited by high light at the coastal station, the ambient rates were large enough that a 50% decline in rates still allowed ~60 nM d$^{-1}$ of ammonia oxidation even when light was increased to 20% surface PAR. The 30 nM d$^{-1}$ difference between DK and HL ammonia oxidation rates at the coastal station was measured over an 8 h incubation, suggesting that the microbial response to light perturbations can be quite fast and large enough to switch NetNO$_2$ from positive to
negative (Fig. 6). However, the observed response to changing light is dependent on the starting community.

Short term light inhibition does not entirely exclude nitrification from the surface ocean. Horak et al. (2018) also tested ammonia oxidation rates of PNM communities under increased light conditions and determined that light was a critical control of ammonia oxidation in the surface ocean, sometimes eliminating ammonia oxidation completely at high light levels. However, the ammonia
oxidation rates measured in the central north Pacific Ocean (<4 nM d$^{-1}$) were lower than the ambient rates in our ETNP light experiments (8-90 nM d$^{-1}$), supporting the idea that initial community is an important determinant of how strongly a light perturbation will affect rates. Our lowest ambient rates were at offshore station PS1 and showed the highest percentage of light inhibition compared to the other stations with higher ammonia oxidation rates (Fig 2i, S5). While high light does partially inhibit rates, ultimately ammonia oxidizers may be excluded from surface oceans due to lack of substrate, low growth rates and/or
sustained light inhibition that occurs over time scales longer than our 8 h incubations.

Changes in phytoplankton activity at the coastal and central stations showed that nitrite release via nitrate reduction has a complicated response to light. We observed increased nitrite release when cells were exposed to both increased light as well as removal of light. Interestingly, release of nitrite by phytoplankton under both low light and high light conditions has been
documented in the literature. Our dataset suggests that both mechanisms may be simultaneously relevant to PNM formation in the ETNP, although ammonia oxidation still dominates nitrite production in the PNM. The increased responsiveness of coastal phytoplankton activity to changes in light confirm that dynamic coastal waters provide opportunity for larger phytoplankton contributions to nitrite cycling. However, the highest rates of nitrate reduction were observed at station PS2, showing that highest nitrite release rates are not always linked to stations with the highest chlorophyll concentration, but rather species composition or
environmental conditions and disturbance history.



## 5 Conclusions

Our experimental data clearly show the influence of light level on both the individual nitrite cycling processes around the PNM feature as well as on net nitrite production rates. Each step of nitrification was independently sensitive to light, with ammonia oxidation having the clearest declining trend with increased light level and nitrite oxidation rate sometimes showing an increase with light level. Nitrification imbalance (NetNit) also declined with light, reflecting the differential responses to light intensity of the two steps of nitrification, with the highest potential for net nitrite production at the lowest light levels. Additionally, based on the difference in rates between dark incubation and low light treatments, nitrification rates near the depth of the PNM are already inhibited by light. Net phytoplankton production of nitrite (NetPhy) was variable across stations and light treatments, showing that phytoplankton can be both net producers and consumers of nitrite under different conditions. In combination, the net response of the whole microbial community varied from net nitrite producing to net nitrite consuming across stations and light levels, but NetNO$_2$ showed a clear declining trend with increases in light for each microbial community tested (Fig. S6d). Ammonia oxidation was a critical nitrite production mechanism at all stations, but we saw evidence of significant contributions from nitrate reduction at central station PS2. While abrupt perturbations in light can influence net nitrite cycling rates, the starting community influences baseline rates and limits response potential. Substrate availability and average light conditions that control microbial abundance and physiology may have more influence on the variance of nitrite concentrations near the PNM, and studies investigating nitrite cycling on longer timescales closer to the nitrite residence time may provide more insight. With the potential for warming and increased stratification of the oceans resulting from climate change, increased stability of the light environment may control the balance of nitrite cycling processes in the primary nitrite maximum.





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

**Author contributions:**

Experimental design, and major data collection efforts, data processing/analysis and writing were conducted by NMT. Significant support during data collection was provided by CLK, with additional contributions during data analyses and manuscript editing. KLC was involved in initial project design, laboratory analysis, data investigations and manuscript writing.

**Acknowledgements:**

Thank you to the Captains and crew of the R/V Sally Ride and the R/V Falkor for assistance and support at sea. Thank you to Chief Scientists Bess Ward (SR1805) and Andrew Babbin (FK180624) for their efforts in support of this work. Thanks also to Anabelle Baya for mass spectrometer assistance during isotope analyses. This work is supported by NSF awards OCE 1657868 and OCE 1736756 to KLC. N. Travis was partly supported by the Stanford Department of Earth System Science. C. Kelly was partly supported by an NSF Graduate Research Fellowship.

**Data availability:**

Cruise data can be accessed on BCO-DMO for SR1805 (https://www.bco-dmo.org/dataset/854091, last access: 1 January 2022, Ward and Casciotti, 2021) and for FK180624 (https://www.bco-dmo.org/dataset/832389, last access: 1 January 2022, Babbin et al., 2021). Corresponding rate measurements can be accessed on the Stanford Digital Repository ( https://purl.stanford.edu/ds821fj1220, Travis et al., 2023).

**Competing interests:**

The contact author has declared that none of the authors has any competing interests.