# Peer review of "Testing the influence of light on nitrite cycling in the eastern tropical North Pacific"

_Biogeosciences, 2023_

## Referee Comment (RC2)

General Comments on the manuscript from Travis et al., Testing the influence of light on nitrite cycling in the eastern tropical North Pacific

In the manuscript *Testing the influence of light on nitrite cycling in the eastern tropical North Pacific,* Travis and colleagues present evidence of light influence in the accumulation of nitrite at the PNM. The manuscript is well written, containing a great set of figures and tables. While I am in support of the paper for publication, I have a few general comments.

General comments

Line 37: There are random numbers at the beginning of the sentence.

Line 39: What does CA mean? I suggest using the full name instead.

Line 67-68: This sentence is a bit unclear, needs revision.

Line 165: There is a misprint in the sentence.

Line 173-174: I am not certain the incubation period used here is enough to determine an actual rate of ammonium oxidation. This different methodology is quite interesting and needs to be explained more in the methods section. I make this point because literature indicates that nitrifying organisms are slow growers, therefore can we be certain that these are actual rates or the rates themselves should be referred to as potential rates in the manuscript.

Line 178: The table seems more like a repeat of the information already written in the methods section, if the authors deem necessary to include the table, I suggest moving the table to the SI document instead.

Line 215: Should first define CV%.

Line 225: Please comment at Line 173-174

Line 259-263: SigmaT should be define what it is, these numbers right now may not mean much to some people.

Figure 7: Some data points are cut-off on the map.

Line 576: nM d-1 needs a superscript.

---

## Author Comment (AC2)

Thank you for your comments and your efforts to improve this manuscript.

General Comments on the manuscript from Travis et al., Testing the influence of light on nitrite cycling in the eastern tropical North Pacific

In the manuscript *Testing the influence of light on nitrite cycling in the eastern tropical North Pacific,* Travis and colleagues present evidence of light influence in the accumulation of nitrite at the PNM. The manuscript is well written, containing a great set of figures and tables. While I am in support of the paper for publication, I have a few general comments.

General comments:
Line 37: There are random numbers at the beginning of the sentence.

Thank you for catching this error. Random numbers will be removed.

Line 39: What does CA mean? I suggest using the full name instead.

California will be written out fully instead of this abbreviation.

Line 67-68: This sentence is a bit unclear, needs revision.

This sentence can be re-written as, "When both nitrate and nitrite are available abundantly available as substrates for phytoplankton, nitrate uptake rates are typically higher than coincident nitrite uptake rates (Collos, 1998)."

Line 165: There is a misprint in the sentence.

Thank you. The citation will be fixed.

Line 173-174: I am not certain the incubation period used here is enough to determine an actual rate of ammonium oxidation. This different methodology is quite interesting and needs to be explained more in the methods section. I make this point because literature indicates that nitrifying organisms are slow growers, therefore can we be certain that these are actual rates or the rates themselves should be referred to as potential rates in the manuscript.

Yes, these rates should be considered potential rates due to the high likelihood that the 15N additions would serve to enhance the rate measurements towards maximum community rates (per volume). Incubations were initially conducted using 8hr 16hr and 24hr lengths, but the 8hr incubation period was selected because it minimizes bottle effects (e.g. substrate depletion and grazer influence) and was long enough to adequately measure changes in $^{15}$N substrate/product. While ammonia oxidizers are slow growers, we intend to measure the rate of

the existing population and minimize rate increases due to population growth. We will make sure to clarify that our measurements are considered potential rates.

Line 178: The table seems more like a repeat of the information already written in the methods section, if the authors deem necessary to include the table, I suggest moving the table to the SI document instead.

We will consider moving the table to the supplement to avoid redundancy.

Line 215: Should first define CV%.

Coefficient of variance will be defined here before use.

Line 225: Please comment at Line 173-174

We will explain the caveats of our measurement method and refer to rates as potential rates.

Line 259-263: SigmaT should be define what it is, these numbers right now may not mean much to some people.

SigmaT will be defined before use.

Figure 7: Some data points are cut-off on the map.

Thank you. Figure 7 will be revised to show all points more clearly .

[Figure]

Line 576: nM d-1 needs a superscript.

This will be fixed and made consistent.

---

## Author Response (AR1)

**Associate Editor Comment:**

**Comment:** My apologies for the long process obtaining two reviews and thank you for submitting to this journal and for responding promptly to referees' feedback. Both are favorable towards eventual publication and I therefore invite you to submit a revised version.

Thank you for considering the reviewer comments and the content of this manuscript. We are appreciative of your time and contribution to improving this work.

**Reviewer 1 Overview:**

**Comment:** This manuscript addresses scientific questions within the scope of Biogeosciences. It tackles the problem of how light influences the relative rates of four microbial processes involved in nitrite cycling in the eastern tropical North Pacific, and as such presents some novel data. The primary nitrite maximum and the base of the euphotic zone was sampled and then, using experimental incubations with 15N labeled substrates, the production of nitrite due to microbial ammonia oxidation and phytoplankton nitrate reduction were measured along with nitrite consumption by nitrite oxidation and nitrite uptake. The conclusion reached were that net nitrite production from these 4 processes was highest in dark treatments and that ammonia oxidation was the dominant process contributing to the net nitrite. The authors say that light may modulate nitrite accumulation in the PNM.

As they describe , historically the nitrite in the PNM has been thought to be due to both phytoplankton that take up and reduce nitrate in the light and then when they sink into the dark release nitrite, and an imbalance of the two steps of microbial nitrification as the nitrite oxidizers are more light-sensitive than the ammonia oxidizers. Apparently, few studies have directly measured the individual steps or processes of nitrite cycling in field collected communities (as is done in this paper) but have inferred the relative rates from microbial cultures. This paper supports the differential responses to light of the 2 steps of nitrification, and dark promoting the highest net nitrite production. This paper says that both microbial and phytoplankton processes occur, but that ammonia oxidation dominates the nitrite cycling ("a critical nitrite production mechanism") and can occur in light up to 25% of surface PAR, although it tends to decrease with light treatments The effect of light on microbial nitrite reduction was not clear-cut and the authors determined that phytoplankton could be both net nitrite producers and consumers, although at one station there were significant contributions from nitrate reduction.

Thank you for your time in reviewing this manuscript. We appreciate your feedback and your efforts to improve this work.

**General Comments**

**Comment:** A few comments and concerns- although the overall presentation is clear and the language fluent, the visuals are extremely hard to read, especially Figure 6 – the different shaded of black and grey ae challenging to discern and I would recommend using maybe colors or patterned approach (e.g. stripes). Fig 6d - lines cannot be discerned. The symbol legends and tic labels are very hard to read as font is so small. The map can only be read if you enlarge the figure on the screen, not much good when as a pdf. Fig. 7- make the symbols larger? And again, the lines are very faded- could these be black instead of grey? I used Tables S1 and S2 a lot when reading the paper so they should be included in the main manuscript. If tight on number of figs and tables, might incorporate Figure 3 with Figure 2- I used both together when reading.

Thank you for your figure suggestions. I have replotted Fig. 6 with higher contrast greyscale and patterned bar plots.

- Figure 6 revised [474]

[Figure]

Figure 7 font and shape sizes have been increased and higher contrast greyscale has been used for the experimental lines. The station colors are now consistent with Figure 6.

- Figure 7 revised [536]

[Figure]

I've left Figure 2 and 3 separate, because combining them onto Figure 2 leads to much smaller individual panels on the multipanel plot and a large white space. The map size will be increased, and Table S1 and S2 will be moved to the main manuscript as Table 2 and Table 3 respectively.

- Map size increased [269]
- Table S1 renamed as Table 1 and moved to main manuscript [271-274]
- Table S2 renamed as Table 2 and moved to main manuscript [276-284]

**Comment:** It would be great to have a small conceptual diagram with the four nitrite cycling processes plus the phytoplankton 15N uptake (Fig 3) to summarize the results, with maybe size of arrows indicating response to light. This diagram could also be used to present a simple mass balance.

Thank you for this suggestion. We have added a conceptual diagram next to the map to visually depict the nitrite cycling processes.

- Added Fig 1 b [269]

[Figure]

**Comment:** I am not a microbial nitrite cycling specialist but one concern I have is where the ammonium comes from in the ETNP to feed the ammonia oxidation. I was hoping that with the emphasis on microbial ammonium oxidation providing the nitrite for accumulation in the PNM, this source would be discussed. The only ammonium data was that in Tables S1 and S2. Maybe I am oversimplifying but at the rates described, the initial nM levels of ammonium available (Tables S1 and S2) would all be gone on the order of hours unless the ammonium was replaced from somewhere else- but from where- grazing?

Thank you for highlighting the importance of source ammonium for these processes. Rates of ammonium regeneration were not directly measured in this study, but other literature measurements suggest rates near the PNM (~1% PAR) on the order of 25-60 nM d$^{-1}$ (Clark et al. 2005, EGU abstract). Dickson and Wheeler (1995) off the Oregon coast measured rates of >400 nM d$^{-1}$ in the surface ocean. In the Atlantic Ocean, Clark et al. (2008) measured ammonium regeneration rates up to 160 nM d$^{-1}$, which was nearly 10x the associated nitrification rates. This suggests that regenerated ammonium can be supplied in excess of loss processes. The range in ammonium regeneration rates from the literature suggest direct measurements in this region would be helpful in better understanding the local turnover. Unfortunately we did not make direct measurements of ammonium regeneration in this study.

Ammonium profiles tend to have many local maxima through the surface layer. Our ammonium data from this cruise is minimal (only 3 discrete profiles, not measured every cast). However the ammonium maxima tend to line up with PNM and Chlorophyll maxima in a predictable vertically stratified pattern, suggesting classic ammonium source from phytoplankton decomposition/grazing. The persistence of the ammonium accumulation below the chlorophyll maxima does hint that any ammonium sources are also fairly persistent and likely have rates either equivalent or greater than the measured consumption processes.

- Ammonium profile data has been added to Figure S1 [Supp Line 10]

[Figure]

**Comment:** I think this mismatch may come from the methodology of using saturating levels of 15N substrate to measure the rates - ammonia is at the nM level and the 15N additions are 10 times the ambient concentrations. This may be lead to an overestimate of the ammonia oxidation being carried out, as these data offer the optimal potential of maximal ammonia oxidation. This is less of a problem for the nitrate and nitrite where ambient levels are uM, so adding 200 nM to measure nitrate reduction is more like adding the trace levels and is more realistic of the ambient situation. I realize the authors describe their rationale for using uniform 200 nM tracer additions and this would not impact the light treatment study as all treatments were given the same. But this approach will likely stimulate rates and overestimate ammonia oxidation relative to the nitrate reduction tracer data that was obtained with trace level tracer additions. This would also explain why in Fig 7 most the experimental ammonium oxidation values are so much higher than the ambient measurements. Then this should be mentioned in the discussion more, and the emphasis on ammonia oxidation relative to phytoplankton nitrate reduction and nitrate uptake put into context.

Yes, you are correct in noting the potential for nitrogen additions to be more/less influential based on the ambient nitrogen concentrations present in the source water for each experiment. Ammonium $^{15}$N additions (200 nM) are often a much larger percentage of the ambient ammonium pool compared to the ambient nitrite and nitrate. Ammonia oxidation kinetics work by Xu. et al (2019) showed that rate measurements in the subtropical western North Pacific were increased 3x with a 20 nM $^{15}$N addition (starting $NH_4$ = 29 nM), with the caveat that initial absolute rates maxed out at 0.48 nM d$^{-1}$ ($V_{max}$) which indicates a significantly different community of ammonia oxidizers than our region of study. Work by Horak et al. (2013) with field communities from the Hood Canal, WA also showed increases in rates up to 6 nM d$^{-1}$ due to 300 nM $^{15}$N spike concentrations (starting $NH_4$ = 50 nM). While the $^{15}$N spike addition doubled the absolute rate, again rates observed in the ETNP region (our study area) can typically be much higher (>20 nM d$^{-1}$).

In work by Beman et al. (2013) from the ETNP region, a uniform 42 nM $^{15}$N spike was used to measure ammonia oxidation rates with ambient ammonium concentration at the ammonium maxima reaching up to 200 nM, and their peak ambient ammonia oxidation rates at each station ranged from ~ 35 to 120 nM d$^{-1}$. These rates are similar in range to the ambient rates measured using our 200 nM $^{15}$N spike methods (this manuscript and Travis et al. 2023), suggesting that the percentage of $^{15}$N added may not influence the variation in rates as much as the variation in archaeal community across stations. However, these rate measurements are both likely to be potential rates (enhanced by the $^{15}$N addition to an unknown degree). While ammonium additions were typically a larger percentage of the ambient ammonium (compared to $^{15}$N-nitrate spikes), since we do not have corresponding kinetics experiments we cannot determine the relative enhancement of each process (e.g. ammonium oxidation vs nitrate reduction). It is likely that each microbial community responds to substrate increases to differing degrees. This caveat will be more clearly noted in the discussion.

In Figure 7, the experimental ammonia oxidation rates for each of the 3 experimental stations span the range of ammonia oxidation rates typically observed at ~1% surface PAR across the region. The high rates observed in the medium and high light experimental treatments cannot necessarily be explained by enhancement from saturating tracer addition levels, since all rates in this figure received 200 nM $^{15}$N-$NH_4$. However, since the ambient $NH_4$ concentrations vary, the degree of enhancement may vary.

- Added [line 706-7710] "Estimation of net community nitrite…. Estimation of net community production of nitrite is dependent on measurement of each contributing process. While a uniform 200 nM of $^{15}$N tracer was used to measure each contributing process, the relative enhancement of each measured rate may not be uniform. Since ambient ammonium concentrations are much lower than nitrate concentrations (nanomolar vs micromolar

levels), the addition of 200 nM of tracer substrate may cause a disproportionately large enhancement in the measured rates of ammonia oxidation. "

**Comment:** Although the paper is focused on the influence of irradiance, the question of where the nitrite in the PNM is always in the background and this emphasis on ammonia oxidation from experimental saturated uptake values may be a bit misleading; the phytoplankton nitrate uptake rates (Fig 3) suggest that phytoplankton may still be important, even if the direct 15NO3 to 15NO2 rates (nitrate reduction) measured with trace isotope do not seem sufficiently high.

Yes, we agree with the nuanced interpretation that nitrate reduction may still be an important contributing process for nitrite production under some conditions. These situations may be slightly obscured by the tendency for our $^{15}$N spikes to enhance measured ammonia oxidation rates more than measured nitrate reduction rates. We will highlight this point more clearly in the discussion.

- Added [line 788- 792] "However, the extent to which the $^{15}$N tracer additions may have enhanced each measured rate is unclear. Thus there is room to improve our understanding of the relative contributions of nitrite coming from ammonia oxidation versus nitrate reduction, and there are likely conditions were nitrate reduction is a significant source of nitrite to the PNM. Assessment of natural abundance isotopes of nitrite may provide further insight into the sources of nitrite in this region (Buchwald and Casciotti, 2013)"

**Comment:** On positive note, the methods and assumptions were clearly outlined, the results both in the supplementary and main body supported their interpretations and the number of references were appropriate. Amount and quality of the supplementary material was appropriate although Figure S1 should be increased in size and a vertical profile of ammonium should be provided for context.

- Figure S1 has been increased in size, and ammonium profiles were added [Supp Line 10]

**Reviewer 2 Overview:**

General Comments on the manuscript from Travis et al., Testing the influence of light on nitrite cycling in the eastern tropical North Pacific

**Comment:** In the manuscript *Testing the influence of light on nitrite cycling in the eastern tropical North Pacific,* Travis and colleagues present evidence of light influence in the accumulation of nitrite at the PNM. The manuscript is well written, containing a great set of figures and tables. While I am in support of the paper for publication, I have a few general comments.

Thank you for your comments and your efforts to improve this manuscript.

**Reviewer 2 Specific Comments:**

**Line 37:** There are random numbers at the beginning of the sentence.

Thank you for catching this error. Random numbers have been removed.

**Line 39:** What does CA mean? I suggest using the full name instead.

California will be written out fully instead of this abbreviation.

**Line 67-68:** This sentence is a bit unclear, needs revision.

This sentence has been re-written for clarity.

- Edited [67-68]: "When both nitrate and nitrite are abundantly available as substrates for phytoplankton, nitrate uptake rates are typically higher than coincident nitrite uptake rates (Collos, 1998)."

**Line 165:** There is a misprint in the sentence.

Thank you. The citation has been fixed.

- Edited [174] (Santoro 2010)

**Line 173-174:** I am not certain the incubation period used here is enough to determine an actual rate of ammonium oxidation. This different methodology is quite interesting and needs to be explained more in the methods section. I make this point because literature indicates that nitrifying organisms are slow growers, therefore can we be certain that these are actual rates or the rates themselves should be referred to as potential rates in the manuscript.

Yes, these rates should be considered potential rates due to the high likelihood that the 15N additions would serve to enhance the rate measurements towards maximum community rates (per

volume). Incubations were initially conducted using 8hr 16hr and 24hr lengths, but the 8hr incubation period was selected because it minimizes bottle effects (e.g. substrate depletion and grazer influence) and was long enough to adequately measure changes in $^{15}$N substrate/product. While ammonia oxidizers are slow growers, we intend to measure the rate of the existing population and minimize rate increases due to population growth. We will make sure to clarify that our measurements are considered potential rates.

- Added [151-153] "Thus, the rates reported may be considered potential rates, especially for ammonia oxidation rates where the $^{15}$N spike is frequently a large percentage of the substrate pool."

**Line 178:** The table seems more like a repeat of the information already written in the methods section, if the authors deem necessary to include the table, I suggest moving the table to the SI document instead.

Good idea. Table 1 has been moved into the supplement.

**Line 215:** Should first define CV%.

Coefficient of variance will be defined here before use.

- Edited [229-230] ",where CV is the coefficient of variance (the ratio of the standard deviation to the mean)."

**Line 225:** Please comment at Line 173-174

We have explained the caveats of our measurement method and refer to rates as potential rates. See above.

**Line 259-263:** SigmaT should be define what it is, these numbers right now may not mean much to some people.

SigmaT is defined at first use.

**Figure 7:** Some data points are cut-off on the map.

- Edited [570] Figure 7 has been revised to show all points more clearly .

[Figure]

**Line 576:** nM d-1 needs a superscript.

Thank you. This will be fixed and made consistent.